# Growth Response of Non-Conventional Yeasts on Sugar-Rich Media: Part 1: High Production of Lipid by *Lipomyces starkeyi* and Citric Acid by *Yarrowia lipolytica*

**DOI:** 10.3390/microorganisms11071863

**Published:** 2023-07-24

**Authors:** Panagiota Diamantopoulou, Dimitris Sarris, Sidoine Sadjeu Tchakouteu, Evangelos Xenopoulos, Seraphim Papanikolaou

**Affiliations:** 1Institute of Technology of Agricultural Products (ITAP), Hellenic Agricultural Organization—Demeter, 1 Sofokli Venizelou Street, Attiki, 14123 Lykovryssi, Greece; pdiamantopoulou@elgo.gr; 2Department of Food Science and Nutrition, School of Environment, University of the Aegean, Metropolite Ioakeim 2, 81400 Myrina, Greece; 3Department of Food Science and Human Nutrition, Agricultural University of Athens, 75 Iera Odos, 11855 Athens, Greece; sadjeu2000@yahoo.fr (S.S.T.); evansxeno@gmail.com (E.X.)

**Keywords:** citric acid, microbial mass, microbial lipids, *Lipomyces starkeyi*, *Yarrowia lipolytica*

## Abstract

Sugar-rich waste streams, generated in very high quantities worldwide, constitute an important source of environmental pollution. Their eco-friendly conversions into a plethora of added-value compounds through the use of microbial fermentations is currently a very “hot” scientific topic. The aim of this study, was to assess the potential of single cell oil (SCO), microbial mass and citric acid (CA) production by non-conventional yeast strains growing on expired (“waste”) glucose. Six yeast strains (*viz*. *Rhodosporidium toruloides* DSM 4444, *Rhodotorula glutinis* NRRL YB-252, *R. toruloides* NRRL Y-27012, *Yarrowia lipolytica* LFMB Y-20, *Y. lipolytica* ACA-DC 50109 and *Lipomyces starkeyi* DSM 70296) were initially grown in shake flasks with expired glucose used as substrate under nitrogen limitation, in order to “boost” the cellular metabolism towards the synthesis of SCO and CA, and their growth response was quantitatively evaluated. Initial glucose concentration (Glc_0_) was adjusted at *c*. 50 g/L. Besides *Y. lipolytica*, all other yeast strains produced noticeable SCO quantities [lipid in dry cell weight (DCW) ranging from 25.3% *w*/*w* to 55.1% *w*/*w*]. Lipids of all yeasts contained significant quantities of oleic acid, being perfect candidates for the synthesis of 2nd generation biodiesel. The highest DCW production (=13.6 g/L) was obtained by *L. starkeyi* DSM 70296, while both *Y. lipolytica* strains did not accumulate noticeable lipid quantities, but produced non-negligible CA amounts. The most promising CA-producing strain, namely *Y. lipolytica* ACA-DC 50109 was further studied in stirred-tank bioreactor systems, while the very promising DCW- and SCO-producing *L. starkeyi* DSM 70296 was further studied in shake flasks. Both strains were grown on media presenting higher Glc_0_ concentrations and the same initial nitrogen quantity as previously. Indeed, *L. starkeyi* grown at Glc_0_ = 85 g/L, produced DCW_max_ = 34.0 g/L, that contained lipid =34.1% *w*/*w* (thus SCO was =11.6 g/L). The strain ACA-DC 50109 in stirred tank bioreactor with Glc_0_ ≈ 105 g/L produced CA up to 46 g/L (yield of CA produced on glucose consumed; Y_CA/Glc_ ≈ 0.45 g/g). Finally, in fed-batch bioreactor experiment, the significant CA quantity of 82.0 g/L (Y_CA/Glc_ = 0.50 g/g) was recorded. Concluding, “waste” glucose proved to be a suitable substrate for a number of non-conventional yeast strains. *Y. lipolytica* ACA-DC 50109 produced significant quantities of CA while *L. starkeyi* DSM 70296 was a very interesting DCW- and SCO-producing candidate. These strains can be used as potential cell factories amenable to convert glucose-based residues into the mentioned metabolic compounds, that present high importance for food, chemical and biofuel facilities.

## 1. Introduction

The concept of valorization is relatively new in the field of industrial residues management, which aims to promote the principle of sustainable development. The objective of valorization of food processing and/or agro-industrial by-products and waste streams, is either based on the recovery of fine chemicals, or the production of (high) added-value metabolites though the implementation of chemical and biotechnological processes [1,2,3]. Given that the various food processes used in the food and drink industry globally generate food supply chain wastes that are difficult to be treated on a multi t. scale every year, valorization of these residues together with production of potentially high-added value lipid could increase the viability of the process being simultaneously beneficial for the environment [1,3]. 

Sugar-rich waste streams are produced in a very high number of industrial sectors, including but not limited to the categories of facilities of manufacture of bread, fresh pastry goods and cakes, manufacture of biscuits and preserved pastry goods/cakes and food preparations for infants [4]. These wastes constitute one of the most important carbon and money losses of the above-mentioned facilities [5]. Moreover, very high quantities of solid simple waste sugars (e.g. waste sucrose or waste glucose employed in the confectionary industries), waste-waters containing high concentrations of sugars (i.e., glucose, fructose, etc.) and, thus, presenting very high levels of COD and BOD values (these waste-waters, therefore, cannot be treated in the typical sewage sludge stations of the aforementioned facilities) or solid residues like waste breads that after hydrolysis generate hydrolysates containing significant quantities of glucose, are annually produced in the relevant food-processing plants [1,5]. Among the seven most important food-processing products in terms of production capacities within the EU27, in three ones (namely grain mill products; sugars and relevant products; bread, fresh pastry goods and cakes) very high quantities of the above-mentioned solid and liquid wastes are generated [1]. The majority of these waste streams are currently utilized as animal feed, as fertilizers and as substrates for composting or vermi-composting processes, while portions of these waste and by-product streams are also land-filled [5]. It is also noted that the removal of a non-toxic and non-hazardous waste materials from the food industry, costs around 0.4–0.7 US$ per kg of waste [1], thus, it can be easily understood that the benefit for the food-processing plant would be very important in relation to the development of cost-effective processes associated with the beneficial conversion of these residues and wastes in situ.

In the present study, the biochemical potential of non-conventional (Crabtree- and Pasteur-negative; *viz*. oxidative) yeast strains cultured on glucose-based media under nitrogen-limited conditions was assessed. Expired (“waste”) solid commercial glucose was employed as substrate. Nitrogen limitation was used in order to “boost” the cellular metabolism towards the synthesis of non-growth coupled metabolic compounds like microbial lipids (single-cell oils; SCOs) and citric acid [1,2,6]. Under nitrogen-limited conditions, in a number of non-conventional yeast (i.e., *Lipomyces starkeyi*, *Rhodosporidium toruloides, Yarrowia lipolytica*, etc.) and filamentous fungal (i.e., *Aspergillus* sp., *Mortierella isabellina*, etc.) species/genera, growth on glucose leads to the de novo accumulation of microbial lipids, to the synthesis of polysaccharides or the secretion of citric acid. The aim of the current submission was to perform physiological and kinetic investigations on non-conventional yeast strains (six strains belonging to the species *R. toruloides, Rhodotorula glutinis, Y. lipolytica* and *L. starkeyi*) on expired glucose employed as sole substrate in nitrogen-limited shake-flask experiments. The most potential citric acid- and lipid-producing strains (*viz. Y. lipolytica* ACA-DC 50109 and *L. starkeyi* DSM 70296) were further studied in more depth. Biochemical and technological considerations of the yeast metabolism were assessed and critically discussed. 

## 2. Materials and Methods

### 2.1. Microorganism and Media

The strains used in this study were the following ones: *Y. lipolytica* ACA-DC 50109, *Y. lipolytica* LFMB Y-20, *R. toruloides* DSM 4444, *R. toruloides* NRRL Y-27012, *R. glutinis* NRRL YB-252 and *L. starkeyi* DSM 70296. Strains with the code characteristics ACA-DC and LFMB Y were provided by the culture collections of the Department of Food Science and Human Nutrition (Agricultural University of Athens, Athens, Greece), strains with the code characteristic NRRL were provided by the NRRL culture collection (Peoria, USA) and strains with the code characteristic DSM were provided by the DSMZ culture collection (Leibniz, Germany). All strains were maintained on yeast/peptone/dextrose agar (YPDA) at *T* = 4 °C and sub-cultured every 3 months in order to maintain their viability. The synthetic medium used had the following salt composition (g/L) [7]: KH_2_PO_4_, 7.0; Na_2_HPO_4_, 2.5; MgSO_4_·7H_2_O, 1.5; CaCl_2_, 0.15; FeCl_3_·6H_2_O, 0.15; ZnSO_4_·7H_2_O, 0.02; MnSO_4_·H_2_O, 0.06. Nitrogen-limited culture conditions were employed, in which peptone and yeast extract were used as nitrogen sources in concentrations of 0.75 g/L and 0.5 g/L respectively. The peptone contained *c.* 18%, *w*/*w* nitrogen and *c.* 30%, *w*/*w* carbon whereas the yeast extract contained *c.* 14%, *w*/*w* nitrogen and *c*. 12%, *w*/*w* carbon. Commercial glucose, the main industrial low-value material utilized in confectionary industries (Hellenic Sugar Industry SA, Thessaloniki, Greece) having ≈90% *w*/*w* purity [impurities composed of maltose (4%, *w*/*w*), malto-dextrines (1%, *w*/*w*), water (4%, *w*/*w*) and salts (1%, *w*/*w*)], that was had been expired, was used as the main carbon source in the various experiments performed. The initial concentration of glucose (Glc_0_) in the shake-flask fermentations carried out was ≈50 g/L (initial molar ration employed at *c.* 115 moles/moles) in order to favor the accumulation of storage lipids and (potentially for the case of *Y. lipolytica*) the secretion of secondary metabolites (i.e., citric acid) [2,6,8,9]. Significantly higher Glc_0_ concentration with the same initial nitrogen quantity (Glc_0_ ranging between 85 and 110 g/L) was employed for batch-bioreactor experiments with use of *Y. lipolytica* ACA-DC 50109 or for shake-flask experiments with *L. starkeyi* DSM 70296 in order to enhance the production of citric acid and/or lipids. The initial pH for all media after sterilization (*T* = 115 °C/20 min for the flasks, *T* = 115 °C/45 min for the bioreactor) was 6.0 ± 0.2. 

### 2.2. Culture Conditions

Prior to any fermentation, the strains were regenerated so as to have an inoculum of three days old. Following, 250-mL Erlenmeyer flasks (pre-culture) filled with 50 ± 1 mL of mineral salts medium (see previously) and containing glucose at 10 g/L, yeast extract at 0.50 g/L and peptone at 0.75 g/L were sterilized and after cooling were aseptically inoculated from the principal freshly regenerated strain and were incubated in orbital shaker (Zhicheng ZHWY 211C; Shanghai, China) for 24 ± 2 hours at 180 ± 5 rpm and *T* = 28 ± 1 °C. Microscopic observation of the yeasts was carried out in order to verify the purity of the strain. Finally, the inoculation of the main culture from the pre-culture took place. Shake-flask experiments of the principal culture were conducted in 250-mL Erlenmeyer flasks, containing 50 ± 1 mL of growth medium and inoculated with 1 mL of a 24-h exponential pre-culture (*c*. 2.0×10^6^ cells, initial biomass concentration at the flasks ≈0.12 g/L). As in the case of the pre-cultures, principal flask cultures were performed in an orbital shaker (Zhicheng ZHWY 211C; Shanghai, China) at 185 ± 5 rpm and incubation temperature *T* = 28 ± 1 °C. In all flask experiments it was desirable to maintain a medium pH in a value greater than 4.8, therefore an appropriate volume of KOH (5 M) was periodically and aseptically added into the flasks [7].

Batch fermentations were also conducted in a laboratory scale bioreactor (apparatus New Brunswick Scientific Co., Edison, NJ, USA), with total volume 3.0 L and working volume 2.0 L, fitted with four probes and two six-bladed turbines. The culture vessel was inoculated with 100 mL (5.0% *v*/*v* inoculum) of exponential pre-culture (see composition of the pre-culture above). The incubation temperature was controlled automatically at *T* = 28 ± 1 °C. Agitation rate was adjusted to 450 ± 10 rpm. A cascade aeration (0.2–2.5 vvm) was employed in order to maintain the dissolved oxygen tension (DOT) at values ≥20% *v*/*v* of saturation, that indicates full aerobic conditions in the performed trial [10]. The pH was automatically controlled at the desired value by adding base quantities of 5 M KOH. All trials were conducted in duplicate and each experimental point of the kinetics was the mean value of two independent determinations.

### 2.3. Analytical Methods

Cells from the whole content of the flasks (≈50 mL) or a content of ≈20 mL from the bioreactor were collected by centrifugation (9000× *g*/15 min at *T* = 10 °C) in a Hettich Universal 320R (Vlotho, Germany) centrifuge and washed twice with distilled water. Biomass (X, g/L) was determined by means of total dry cell weight (DCW). Total wet biomass was put in pre-weighted McCartney flasks and the whole was left at *T* = 90 ± 2 °C/26–28 h. In the flask experiments, a Jenway 3020 pH-meter (Cole-Parmer, Eaton Socon, UK) was used for the off-line pH-measurements. pH was always kept within the range of 4.8–5.8, by adding (periodically and aseptically) small quantities (e.g., 500–600 μL) of 5 M KOH into the flasks when it was required (mostly in the case of the trials with *Y. lipolytica*, since citric acid quantities were accumulated into the culture medium) [7]. As far as the other non-conventional yeast species/strains were concerned, the pH of the culture medium remained almost constant at a value of 5.5 ± 0.3, therefore no need for the correction of pH was required. Dissolved oxygen tension (DOT) was off-line determined using a selective electrode (OXI 96, B-SET, Germany) as described in Filippousi et al [11]. Oxygen saturation was for all strains and all culture phases ≥20% (*v*/*v*). 

Compounds found into the medium, i.e., non-consumed glucose (Glc), mannitol and citric acid (CA), were quantitatively determined via HPLC analysis as described in Diamantopoulou et al [12]. Due to not very satisfactory separation of citric acid from *iso*-citrate, *iso*-citric was also assayed according to Papanikolaou et al [7]. *Iso*-citric acid represented of 7–11% *w*/*w* of the produced CA. Throughout the current submission, in the text and the figures, the concentration of CA was shown. Free amino-nitrogen (FAN) concentration into the fermentation medium (in mg/L) was determined as in Kachrimanidou et al [13].

Total cellular lipids (L, expressed as g/L and % of DCW) for all microorganisms except *L. starkeyi*, were extracted from DCW according to the conventional procedure implicating the utilization of chloroform/methanol (C/M) 2/1 (*v*/*v*) blend, as proposed in Sarantou et al [14]. For the case of the microorganism *L. starkeyi*, digestion with concentrated HCl was performed, as exactly was presented in Sarantou et al [14]. Specifically, a precisely weighted yeast total DCW quantity (up to 300 mg), previously put in a McCartney vial, was acidified by 4.0 mL HCl (2.0 M), at *T* = 80 °C for 60 min. Then the vials were left to cool, C/M 2:1 (*v*/*v*) mixture (up to 20 mL) was added into the vials, and after slight stirring, they were hermetically closed and left for 24 hours in the dark. Thereafter, the lower organic phase (*viz*. chloroform containing cellular lipids) was collected, anhydrous MgSO_4_ was added and, therefore, the existing traces of water were removed, this organic phase was filtered through a Whatman® n° 3 filter paper to remove potential cell debris and precipitates of salts, and finally chloroform phase was collected in pre-weighted evaporator flask, was evaporated in the rotary evaporator as previously mentioned and total lipids were quantitatively determined gravimetrically. In some cases, lipids were fractionated into their lipid fractions over a column of silicic acid as previously mentioned [15]. Cellular lipids or lipid fractions were converted to their corresponding fatty acid methyl-esters (FAMEs) in a two-step reaction [7], and FAMEs were analyzed according to Fakas et al. [15] and were identified by reference to standards. Equally for *L. starkeyi* lipids, crude lipid extract (C/M extract) or neutral lipids were subjected to TLC analysis. The separation and the visualization of the plates was performed according to Papanikolaou et al. [16] with some differences; namely, the separation of crude and neutral lipid was carried out with *n*-hexane/diethyl ether/glacial acetic acid (80:20:1, *v/v/v*), while the visualization was performed to an iodine chamber.

In the case of the trials with *L. starkeyi*, total intra-cellular polysaccharides (IPS, expressed as g/L and % of DCW) were measured based on a modified protocol published by Argyropoulos et al [17]. Briefly, 0.05 g of DCW was acidified by adding 20 mL HCl (2.5 M). The acidified solution was then hydrolyzed at *T* = 100 °C for 30 min and was neutralized to pH 7.0 with KOH (2.5 M), was filtered through Whatman filter paper and was subjected to determination of reducing sugars, according to the 3,5-dinitrosalicylic acid method. 

### 2.4. Data Analysis

Each experimental point of all of the kinetics presented in the tables and figures is the mean value of two independent determinations, where two lots of independent cultures using different inocula were conducted. Standard error (SE) was for most experimental points ≤15%. Data were plotted using Kaleidagraph 4.0 Version 2005 showing the mean values with the standard error mean.

## 3. Results and Discussion

### 3.1. Initial Screening of Yeast Strains on Glucose Base Media

The six employed non-conventional strains were tested on media composed of Glc_0_ concentration ≈50 g/L under nitrogen-limited conditions (utilization of peptone at 0.75 g/L and yeast extract at 0.50 g/L; initial molar ratio employed ≈115 moles/moles) in order to favor the accumulation of storage lipids and (potentially for the employed *Y. lipolytica* strains) the secretion of secondary metabolites (mostly citric acid) useful for the Food Industry. The obtained results of the performed trials as regards biomass and lipid production of the screened strains are illustrated in Table 1.

The yeast *L. starkeyi* DSM 70296 presented the highest total biomass formation (18.0 g/L), which at that fermentation point contained lipid to *c*. 16% *w*/*w* (Table 1). The concomitant dry biomass yield per unit of glucose consumed (Y_X/Glc_) was ≈0.41 g/g. Before achieving its highest DCW quantity, the microorganism had proceeded to non-negligible lipid accumulation in a previous stage (X = 13.8 g/L, lipid in DCW ≈29% *w*/*w*; see Table 1), and thereafter it proceeded to lipid break-down with concomitant rise in its value of total DCW. Lipid biodegradation in order for (mostly) lipid-free material to be produced has been reported in many cases implicating oleaginous microorganisms like *Y. lipolytica* [14,18], *R. toruloides* [19], *Cryptococcus curvatus* [20,21], *Mucor circinelloides* [22], etc, and has been reported to be a process independent of the culture “pre-history” (i.e., the carbon source employed in order for cellular lipids to be synthesized). Furthermore, the higher lipid production per unit of DCW (=55.1% *w*/*w*—see Table 1) was reached by *R. toruloides* DSM 4444. Remarkable biomass production (X > 11 g/L; Y_X/Glc_ ≈ 0.27 g/g) and interesting lipid accumulation was observed for the strains *R. glutinis* NRRL YB-252 and *R. toruloides* NRRL Y-27012. 

What was interesting in the current submission and coincided with previous information concerning the growth of several strains of the species *Y. lipolytica* on hydrophilic carbon sources (e.g., glycerol, glucose, etc.) [14,18,23,24] was the fact that lipid in DCW values in this microorganism, at the first stages of the culture were relatively elevated, despite the fact that nitrogen was found in excess into the growth medium (i.e., 17.1% *w*/*w* for the strain LFMB Y-20 and 18.2% *w*/*w* for the strain ACA-DC 50109). Thereafter, and although significant quantities of glucose remained untouched into the growth medium, lipid in DCW values noticeable decreased (see Table 1), while glucose assimilation led mainly to secretion into the medium of total citric acid. In agreement with the literature [7,24,25,26,27,28,29], during growth of both *Y. lipolytica* strains on glucose, CA was produced almost exclusively at the stationary growth phase, while, as stated, CA biosynthesis and secretion coincided with significant decrease in the quantity of lipids produced per unit of biomass synthesized. In the strain *Y. lipolytica* ACA-DC 50109 the maximum CA quantity was =19.8 g/L (the concomitant conversion yield of CA produced per unit of glucose, Y_CA/Glc_, was ≈0.41 g/g), while in the strain LFMB Y-20 the CA_max_ concentration reported was up to *c*. 14 g/L (yield Y_CA/Glc_ ≈ 0.29 g/g), while also mannitol at 5.8 g/L was produced and secreted into the medium. 

The fatty acid (FA) composition of the cellular lipids produced by the studied yeast strains is presented in Table 2 (analysis performed at the stationary growth phase of the performed trials). In agreement with most reports appeared in the literature (see, i.e., state-of-the-art review-articles like [2,6,9,30] the principal FAs found in variable quantities were mainly the oleic acid (^Δ9^C18:1), and the palmitic acid (C16:0). Cellular lipids of the yeasts *R. toruloides* NRRL Y-27012 and *L. starkeyi* DSM 70296 contained increased concentrations of C16:0 (≥30% *w*/*w* of total lipids), which when added to the concentration of the cellular stearic acid (C18:0) reach values of saturated FAs ≥40% *w*/*w* of total cellular lipids, that are values near to the ones of palm oil [6,31]. The production and subsequent utilization of palm oil (this oil presents enormous applications in the sectors of food and biodiesel production) is one of the leading contributors to tropical deforestation, resulting in habitat destruction and increased CO_2_ emissions [31], therefore potential production in large-scale operations of microbial alternatives of this fatty material (like the lipids of *R. toruloides* NRRL Y-27012 and *L. starkeyi* DSM 70296) would have much to offer to the resolution to the above-mentioned remarkable problems. On the other hand, all yeast lipids produced (see Table 2) contained significant quantities of the FA ^Δ9^C18:1, rendering them as ideal candidates for the synthesis of 2nd generation biodiesel [6,9] and implying the potential of their subsequent utilization as a starting material in chemo-enzymatic syntheses [32], specifically if microbial lipids are produced when low-cost compounds are implicating as starting materials of the bioprocesses. Poly-unsaturated cellular FAs were not detected in high concentrations. Linoleic acid (^Δ9,12^C18:2) was the only poly-unsaturated FA detected in somehow elevated quantities (up to *c*. 20% *w*/*w* of total lipids) only for the strains of *Y. lipolytica*, whereas other yeasts presented this FA in low or indeed minimal quantities (Table 2). The presence of ^Δ9,12^C18:2 in the lipids of *Y. lipolytica* is in accordance with the observations done for several wild-type *Y. lipolytica* strains cultivated on glucose-based media or similarly catabolized compounds (e.g., glycerol) under nitrogen-limited conditions [12,18,24,25]. In general, the poly-unsaturated cellular FAs and specifically the ones that contain into their aliphatic chain ≥3 double bonds, are the principal storage lipophilic compounds in oleaginous fungi and algae [30,33], and, in general, they can be produced in significant quantities inside the yeast cells only after appropriate genetic modifications [30].

### 3.2. Culture of L. starkeyi at Higher Initial Glucose Media

Given that *L. starkeyi* DSM 70296 presented noticeable DCW production, this microorganism was further studied at higher Glc_0_ concentrations (=85 g/L) in which the initial nitrogen availability into the medium remained as previously (the initial concentrations of yeast extract and peptone were 0.50 and 0.75 g/L respectively). Therefore, besides the increased Glc_0_ concentration into the medium, significantly higher initial molar ratio C/N was employed (in the latter case, the initial C/N molar ratio was ≈195 moles/moles). Compared therefore to the previous set of trials (see Table 1), noticeably higher nitrogen limitation was imposed, a fact that would be attainable to “boost” the cellular metabolism towards the synthesis of higher quantities of cellular lipids [2,6]. Indeed, significant quantities of total DCW and total lipid were produced; 216 h after inoculation, the microorganism was reported to produce the significant total DCW quantity of 34.0 g/L that contained 34.1% *w*/*w* of lipids in DCW, therefore 11.6 g/L of lipids were produced. It is interesting to indicate that glucose assimilation and total DCW production seemed both to be almost linear as function of the fermentation time (Figure 1a). Also, despite nitrogen excess conditions and balanced growth that occurred at the first fermentation steps (balanced phase occurred up to t = 50 h after inoculation; the initial FAN concentration was ≈50 mg/L, and at time t > 50 h, this concentration became =10 ± 4 mg/L, demonstrating that after that point nitrogen-limited conditions occurred into the medium), non-negligible quantities of intra-cellular polysaccharides had been detected (IPS/X ≈ 25% *w*/*w*) (Figure 1b). At the same fermentation points, lipids in total DCW remained at low levels (at that fermentation period L/X values were <11% *w*/*w*), typically significantly increasing to vales >30% *w*/*w* after nitrogen limitation (Figure 1b). 

In the de novo lipid biosynthesis process, lipid accumulation is initiated, mostly, after exhaustion of nitrogen from the medium. Nitrogen exhaustion leads to a rapid decrease of the concentration of cellular AMP, which is further cleaved in order for nitrogen to be offered to the cells. Cellular AMP concentration decrease alters the Krebs cycle function resulting in the accumulation of intra-mitochondrial citric acid. When the concentration of citric acid inside the mitochodria becomes higher than a critical value, it is secreted inside the cytoplasm. Then, citric acid is cleaved by ATP-citrate lyase, enzyme-key showing the oleaginous character of the microorganisms, into acetyl-CoA and oxaloacetate, and acetyl-CoA, through the action of fatty acid synthase generates cellular fatty acids and subsequently triacylglycerols (TAGs) (for reviews see: [2,6,9,31,34]). Intra-cellular polysaccharides theoretically are accumulated inside the yeast cells through similar biochemical mechanisms with these of cellular lipids (nitrogen limitation can, in addition to lipid accumulation, also trigger biosynthesis and the production of endopolysaccharides [30,33,34], therefore the non-negligible accumulation of polysaccharides at the early growth steps seems as a non-expected kinetic and physiological result. Nevertheless, it must be pointed out that for several species of oleaginous yeasts (mostly belonging to *R. toruloides* and *C. curvatus*) cultivated under conditions enabling the de novo lipid accumulation process (i.e., batch sugar- or glycerol-based nitrogen-limited ones), similar physiological behavior with increased concentrations of endopolysaccharides at the early growth phases has been reported, despite the nitrogen presence into the medium [12,14,32,35]. Moreover, the conversion yield of total DCW produced per unit of glucose consumed was constant throughout the culture presenting quite high values (Y_X/Glc_ = 0.39 g/g) (Figure 1c), while the conversion yield of lipid produced per unit of glucose consumed was constant with a Y_L/Glc_ value being =0.14 g/g (Figure 1d). 

The analysis of cellular lipid produced by *L. starkeyi* (t = 180 h after inoculation; at that time total DCW was 21.5 g/L and L was 6.7 g/L) is demonstrated in Figure 2. According to the TLC analysis performed, neutral lipids contained mainly triacylglycerols and sterols while almost no quantities at all of steryl-esters were identified (in fact, fungal lipids cannot contain cholesterol, that was the sterol used for identification purposes; most probably this spot should correspond to ergosterol). Total cellular lipids contained mainly triacyglycerols, sterols, potentially some phospholipids (PLs) quantities (that were not sufficiently separated from sterols), while low quantities of glycolipids + sphingolipids (G + S) were identified. Additionally, another unknown compound was identified within total *L. starkeyi* lipids (Figure 2). Gravimetric determination of the lipid fractions (i.e., neutral lipids; NLs; G + S; PLs) and FA composition analysis of the mentioned fractions is illustrated in Table 3. From the performed analysis it can be indicated that, by far the most abundant class of the microbial lipids produced was that of the NLs (mostly composed, as previously demonstrated by triaclyglycerols). The FA composition of individual cellular lipid fractions demonstrated that the cellular PLs was by far the most unsaturated amongst lipid fractions analyzed, in accordance with the results reported for other oleaginous and non-oleaginous microorganisms like *Cunninhgamella echinulata* [15], *Mortierella isabellina* [36] and (various strains of) *Y. lipolytica* [18,25]. By contrast, detailed lipid analysis demonstrated that the cellular PLs were identified as by far the most saturated fraction amongst lipid fractions of the mycelia of the edible fungus *Volvariella volvacea* when flask-cultured on glucose-based media [37]. Moreover, total FA composition analysis of the cellular lipids of *L. starkeyi* in various fermentation points, demonstrated that in all instances the composition did not significantly change (Table 4), no correlation concerning the various (small) changes in the FA composition of the cellular lipids as function of time could be established, while in general, due to the significant concentration of the cellular palmitic and oleic acid that was observed in all fermentation points, yeast lipid presented FA composition similarities with these of palm oil [6,31].

The quantities of total DCW and lipid achieved in the current investigation (X = 34.0 g/L that contained 34.1% *w*/*w* of lipids in DCW, therefore L was =11.6 g/L) are quite impressive ones and compare favorably with many studies that have been carried out, specifically in shake-flask trials. Total DCW production (g/L) and concomitant lipid in DCW (%, *w*/*w*) values achieved by various *L. starkeyi* strains growing in several types of substrates appear in Table 5. It is interesting to indicate that indeed significant DCW values (>80 g/L) have been obtained by the strain under investigation (DSM 70296) during growth on various food waste hydrolysates in fed-batch bioreactor experiments [38,39]. These mentioned literature reports together with the promising results achieved in the present study, demonstrate the dynamics of the strain in relation to its potential application using sugar-based residues or renewable resources as microbial substrates, in order to produce palm oil equivalents in semi pilot-scale operations. 

### 3.3. Citric Acid Production by Y. lipolytica in Bioreactor Experiments

Given that *Y. lipolytica* ACA-DC 50109 presented non-negligible production of CA in the shake-flask experiments, it was decided to proceed to further cultivation of this strains on glucose-based experiments under higher Glc_0_ concentrations, in bioreactor trials. It should be indicated that in earlier or more recent reports [12,52] the mentioned strain has been revealed capable to produce some non-negligible quantities of CA quantities (i.e., up to 62.5 g/L, volumetric productivity 0.10 g/L/h) during growth on glycerol, but in all cases these results were achieved in shake-flask trials. Therefore, bioreactor experiments were employed using expired (“waste”) glucose, in order to increase both the final CA concentration and volumetric productivity. Indeed, the strain was cultured at media with Glc_0_ ≈ 105 g/L, with initial nitrogen concentration added as peptone and yeast extract at concentrations 0.75 g/L and 0.5 g/L respectively (the same as in the previous shake-flask trials), and when glucose assimilation was significantly reduced (at t = 85 h after inoculation), a concentrated solution of expired glucose was added into the bioreactor (therefore, a fed-batch trial with intermittent feeding was performed (see Figure 3a,b). Glucose assimilation rate as calculated by the formula $r_{Glc}=-\frac{\Delta Glc}{\Delta t}$, for the first stage of the culture (0–85 h) was ≈1.2 g/L/h. At that time (t = 85 h) the achieved CA concentration was =46.0 g/L, and for this stage (batch) of the culture, the CA conversion yield on glucose consumed (Y_CA/Glc_) was ≈0.45 g/g. Thereafter, a small quantity of concentrated glucose solution was added into the medium (at t = 85 h, the addition of concentrated solution resulted in Glc concentration =58.4 g/L into the reactor), and for the second stage of the culture (i.e., until t = 136 h, where all of the quantity of glucose had been assimilated) the assimilation rate of glucose was ≈1.1 g/L/h (a rate that was almost equal with the first stage). 136 h after inoculation, a final CA concentration =82.0 g/L had been achieved, with a corresponding volumetric productivity of 0.60 g/L/h. Typically, at the first culture steps (i.e., at the balanced growth phase, that occurred up to t ≈ 30 h), glucose was mainly consumed in order for total DCW production to be performed, while simultaneously very small CA quantities were detected. Thereafter, while biomass concentration remained grosso modo constant (with X ranging between 7.5 and 8.5 g/L), CA concentration drastically and almost linearly increased, with a final concentration as stated, being =82 g/L. Representation of CA produced per consumed glucose for the whole set of data, revealed quite satisfactorily (*R* = 0.987) the global conversion yield of CA produced per unit of glucose consumed (Y_CA/Glc_) that was =0.50 g/g. 

The CA production value obtained in the bioreactor trial (CA_max_ = 82.0 g/L; Y_CA/Glc_ = 0.50 g/g) compares favorably with most reports in which wild-type *Y. lipolytica* strains have been employed as cell factories, and can be considered as quite interesting, but (slightly or somehow) lower as compared with the CA_max_ values obtained from mutant (mostly acetate or acotinase ones) or genetically engineered strains reported so far in the literature [8]. One of the most important factors influencing the final production of CA during growth of *Y. lipolytica* strains on various carbon sources employed as substrates, is the utilization of wild-type or mutant/genetically modified strains for this purpose [8]. For instance, CA_max_ quantities ranging between 13 and 150 g/L have been reported, with the wild-type strains in most cases synthesizing CA in some tenths g/L (i.e., maximum concentrations up to 50 or 60 g/L are considered very promising from these types of strains), whereas by mutant (mostly acetate-negative mutants) or genetically engineered strains, the corresponding concentrations can be in several cases >100 g/L. Likewise, the maximum values of conversion yields of CA produced per unit of substrate (mostly sugar or glycerol) consumed can be within the range of the values achieved in the current submission (0.40–0.70 g/g; rarely values >0.80 g/g are achieved) [8]. On the other hand, the volumetric productivity of CA achieved in the present study (≈0.60 g/L/h), is quite satisfactory and comparable with the highest values reported in the literature by other strains (wild-type or mutant/genetically engineered ones) appeared in the literature in batch (0.48–0.85 g/L/h; [19,53]) and fed-batch (i.e., 0.60–1.12 g/L/h; [54,55,56,57]) bioreactor systems, with shake-flask experiments achieving much lower values (i.e., 0.10–0.25 g/L/h; [8]). Some comparisons with the literature (earlier or more recent one) concerning CA production are depicted in Table 6.

## 4. Concluding Remarks

Glucose-rich waste streams and related compounds (i.e., waste breads, “expired” sucrose/glucose/sugar syrups, sugar-rich residues deriving from industries of manufacturing of fresh pastry goods, cakes, biscuits, creams, food preparations for infants, etc.) are produced worldwide in very high quantities and their eco-friendly conversion into compounds that could present interest for several industrial sectors is currently one of the “hottest” topics in industrial biotechnology [3,4,5]. In this context, in the present study, expired (“waste”) glucose was employed as sole carbon source in a number of non-conventional yeast strains, with trials being performed under nitrogen limitation, in order to assess the dynamics of SCO and CA production in these strains. Two of the tested strains belonging to the species *Y. lipolytica*, typically, did not accumulate notable lipid quantities but shifted their metabolism towards the synthesis of (mostly) CA. From the remaining strains, *L. starkeyi* DSM 70296 presented the best performances concerning DCW production, and for this reason it was grown at higher Glc_0_ concentration media, and produced a DCW_max_ quantity =34.0 g/L, that contained lipids at 34.1% *w*/*w* (SCO = 11.6 g/L). These lipids, mainly composed of triacylglycerols, presented composition similarities with the palm oil. Also, the *Y. lipolytica* strain ACA-DC 50109, that previously was revealed capable to synthesized CA, when grown in aerated and agitated fed-batch bioreactors, produced noticeable CA quantities (CA = 82 g/L, yield Y_CA/Glc_ = 0.50 g/g). The current study therefore, provides strong evidence that the two previously studied wild-type non-conventional yeasts can successfully be used as model microbial cell factories that are amenable to convert expired (“waste”) glucose or other glucose-based and glucose-rich residues (i.e., waste glucose syrups, hydrolyzed waste bread, hydrolyzed solid residues deriving from cake-, biscuit-, and cream-manufacturing processes, etc.) into added-value metabolic compounds of importance of the food, the pharmaceutical, the chemical and the biofuel industries; specifically, from “waste” glucose, *L. starkeyi* DSM 70296 produced high quantities of lipids presenting composition similarities with the palm oil. This finding presents importance, given that the enormous utilization of palm oil in the sectors of food and biodiesel production is one of the most important reasons to the tropical deforestation and the subsequent increased CO_2_ emissions occurring worldwide. On the other hand, *Y. lipolytica* ACA-DC 50109 produced from “waste” glucose high quantities of CA, a compound with very important and numerous applications in the food and beverage industries, in cosmetics and pharmaceutical applications and as dietary supplement, and cleaning/chelating agent. To conclude, expired glucose was successfully converted with the aid of microbial fermentations into added-value microbial compounds with various and important industrial applications. 

## Figures and Tables

**Figure 1 microorganisms-11-01863-f001:**
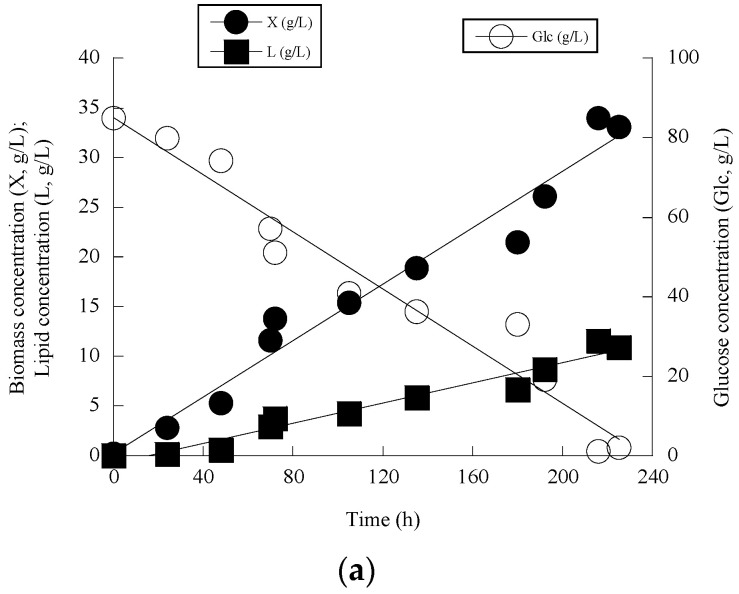
Changes of biomass (X, g/L), glucose (Glc, g/L), and lipid (L, g/L) (**a**) and lipid in DCW and endopolysaccharides in DCW (**b**) as function of fermentation time for *Lipomyces starkeyi* grown on glucose in shake-flask trials. Representation of global conversion yield of biomass produced per unit of glucose consumed as shown by linear regression of produced biomass as function of consumed glucose (**c**) and representation of global conversion yield of lipid produced per unit of glucose consumed as shown by linear regression of produced lipid as function of consumed glucose (**d**) for the same set of data. Each point is the mean value of two independent measurements. SE ≤ 15%.

**Figure 2 microorganisms-11-01863-f002:**
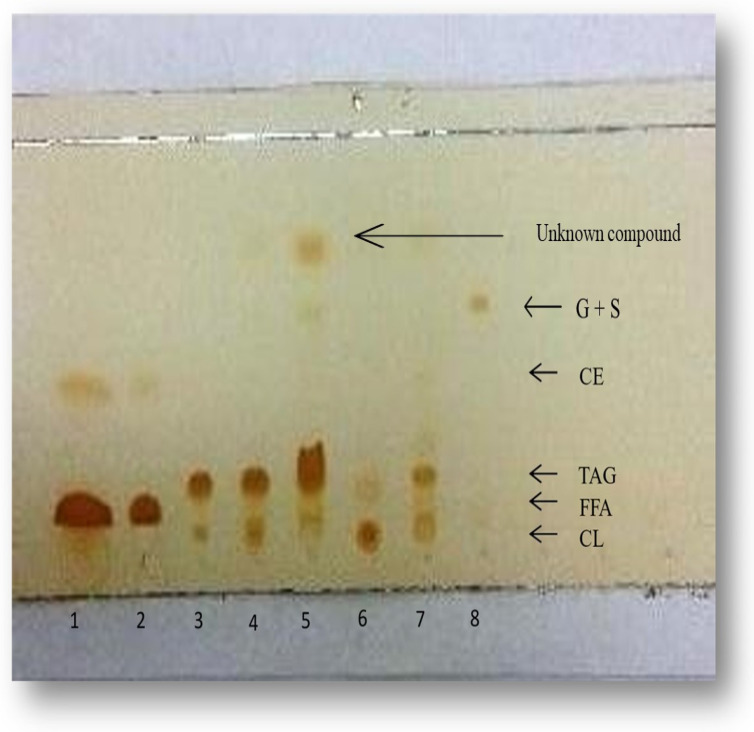
TLC analysis of the “crude lipid” extract at 40 mg/mL (lane 5), neutral lipid at 20 mg/mL (lane 3) and neutral lipid at 30 mg/mL (lane 4) of the cellular lipid of *Lipomyces starkeyi* during growth in nitrogen-limited media containing expired glucose (Glc_0_ = 85 g/L). Dilutions of all lipids were performed on chloroform. Lane 1: oleic acid diluted on chloroform (20 mg/mL); Lane 2: oleic acid diluted on chloroform (10 mg/mL); Lane 6: phospholipid working solution diluted on chloroform (containing L-α-Phosphatidyl-L-serine, L-α-Phosphatidylocholine and 3-sn-Phosphatidylethanolamine); Lane 7: working solution containing cholesteryl linoleate (CE), trioleine (triacylglycerol, TAG) and cholesterol (CL); Lane 8: sphingolipids + glycolipids working solution (containing sphingomyelin and diglycosyldiacylglycerol).

**Figure 3 microorganisms-11-01863-f003:**
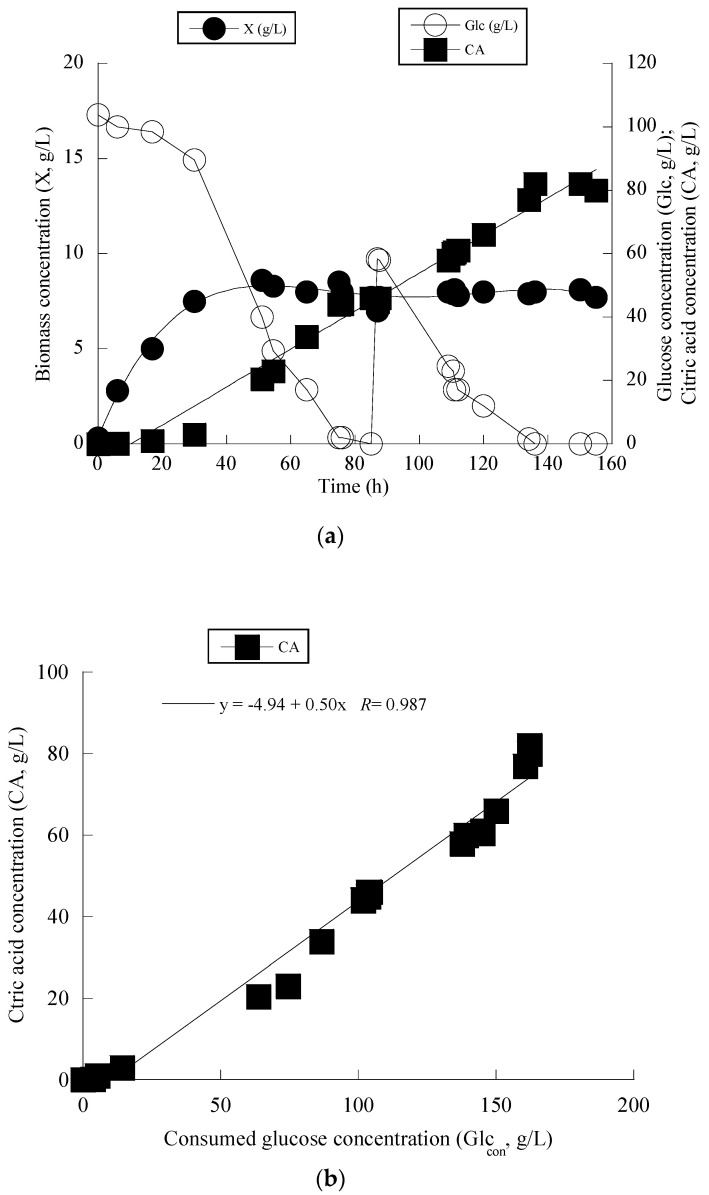
Changes of biomass (X, g/L), glucose (Glc, g/L), and citric acid (CA, g/L) (**a**) as function of fermentation time for *Yarrowia lipolytica* grown on glucose in aerated and agitated fed-batch bioreactor experiments. Representation of global conversion yield of citric acid produced per unit of glucose consumed as shown by linear regression of produced citric acid as function of consumed glucose (**b**). Each point is the mean value of two independent measurements. SE ≤ 15%.

**Table 1 microorganisms-11-01863-t001:** Experimental results originated from kinetics of yeast strains grown on expired solid glucose in shake-flask experiments.

Strain	Point	Time (h)	Glc_cons_ (g/L)	X (g/L)	L (g/L)	Lipid in DCW (%, *w*/*w*)
*R. toruloides* DSM 4444	a, b	168	48.8 ± 2.1	8.9 ± 0.9	4.9 ± 0.5	55.1
*R. glutinis* NRRL YB-252	a, b	192	45.0 ± 2.2	11.9 ± 1.0	3.0 ± 0.4	25.3
*R. toruloides* NRRL Y-27012	a, b	168	40.0 ± 3.6	11.4 ± 1.8	4.5 ± 0.7	39.5
*L. starkeyi* DSM 70296	a	72	33.9 ± 2.2	13.8 ± 1.9	4.0 ± 0.6	28.9
	b	168	44.1 ± 2.9	18.0 ± 2.0	2.9 ± 0.9	16.1
*Y. lipolytica* LFMB Y-20	a	24	7.1 ± 1.7	4.1 ± 0.9	0.7 ± 0.2	17.1
	b	195	48.1 ± 2.1	7.4 ± 1.8	0.6 ± 0.2	8.1
*Y. lipolytica* ACA-DC 50109	a	26	8.2 ± 1.5	4.4 ± 0.8	0.8 ± 0.3	18.2
	b	212	48.8 ± 0.9	6.4 ± 1.3	0.6 ± 0.3	9.4

Representations of biomass (X, g/L), lipid (L, g/L), glucose consumed (Glc_cons_, g/L), fermentation time (h) and lipid in dry biomass (%, *w*/*w*) when the maximum quantity of lipids in dry cell weight (%, *w*/*w*) (a) and the maximum concentration of biomass (X, g/L) (b) was obtained. All cultures were performed on 250-mL flasks. Culture conditions as in “Materials and Methods. Each experimental point is the mean value of two determinations. Standard error (SE) ≤ 15%.

**Table 2 microorganisms-11-01863-t002:** Fatty acid composition of the cellular lipids produced by yeast strains cultivated on expired solid glucose in shake-flask experiments (Glc_0_ ≈ 50 g/L).

Yeast strain	C16:0	C18:0	^Δ9^C18:1	^Δ9,12^C18:2
*R. toruloides* DSM 4444	22.5 ± 2.1	8.5 ± 2.5	57.1 ± 4.1	7.4 ± 1.9
*R. glutinis* NRRL YB-252	18.8 ± 2.8	2.1 ± 0.6	59.9 ± 5.5	2.1 ± 0.4
*R. toruloides* NRRL Y-27012	30.4 ± 3.9	12.9 ± 2.1	50.7 ± 4.1	4.9 ± 0.9
*L. starkeyi* DSM 70296	36.6 ± 3.6	5.4 ± 1.1	50.1 ± 5.1	1.6 ± 0.4
*Y. lipolytica* LFMB Y-20	16.1 ± 1.9	7.3 ± 0.6	44.5 ± 2.6	18.7 ± 2.4
*Y. lipolytica* ACA-DC 50109	14.9 ± 1.9	8.8 ± 1.6	46.9 ± 4.4	19.9 ± 2.6

Time of fermentation for the determination of the fatty acid composition was between 150 and 200 h after inoculation. Each experimental point is the mean value of two determinations. SE ≤ 15%. Culture conditions as in the Table 1.

**Table 3 microorganisms-11-01863-t003:** Distribution of lipid fractions and fatty acid composition of total lipids (C/M 2/1 *v*/*v* extract), neutral lipids (NLs), glycolipids + sphingolipids (G + S) and phospholipids (PLs) of *Lipomyces starkeyi* DSM 70296 during growth in nitrogen-limited media containing expired glucose (Glc_0_ = 85 g/L).

	Quantity (%, *w*/*w*)	C16:0	C18:0	^Δ9^C18:1	^Δ9,12^C18:2
Total lipids (C/M 2/1 *v*/*v*)		34.5 ± 4.1	7.1 ± 1.1	52.4 ± 5.2	1.8 ± 1.1
NLs	89.6 ± 2.1	42.1 ± 4.9	8.1 ± 1.8	47.9 ± 4.4	1.6 ± 0.8
G + S	5.5 ± 1.9	39.8 ± 2.8	7.7 ± 1.9	48.1 ± 5.5	1.7 ± 0.5
PLs	4.9 ± 1.6	26.8 ± 2.5	4.4 ± 1.8	58.9 ± 3.1	3.0 ± 1.1

Culture conditions as in Table 1, sampling point in which lipids were analyzed was at t = 180 h after inoculation. SE ≤ 15%.

**Table 4 microorganisms-11-01863-t004:** Distribution of cellular fatty acids of total lipids of *Lipomyces starkeyi* DSM 70296 during growth in nitrogen-limited media containing expired glucose (Glc_0_ = 85 g/L) during various fermentation times.

Fermentation Time (h)	C16:0	C18:0	^Δ9^C18:1	^Δ9,12^C18:2
48	34.1 ± 3.1	6.5 ± 2.1	47.1 ± 4.1	2.4 ± 0.9
72	37.2 ± 2.5	7.9 ± 3.0	49.9 ± 3.5	2.1 ± 0.4
135	40.9 ± 2.7	5.5 ± 2.1	51.7 ± 4.1	1.9 ± 0.5
192	38.1 ± 3.9	6.6 ± 2.8	52.1 ± 5.3	1.6 ± 0.4

Culture conditions as in Table 1. SE ≤ 15%.

**Table 5 microorganisms-11-01863-t005:** Total biomass and lipid production by *Lipomyces starkeyi* strains growing on various carbon sources and fermentation configurations, and comparisons with the present investigation.

Strain	Culture Type	Substrate	X (g/L)	Lipid in DCW (%, *w*/*w*)	Reference
DSM 70295	Batch shake flasks	Sewage sludge/glucose	9.4	68.0	Angerbauer et al. [40]
AS 2.1560	Batch shake flasks	Glucose/xylose blend	20.5	61.5	Zhao et al. [41]
NRRL Y-11557	Batch shake flasks	Molasses	8.4	14.6	El-Naggar et al. [42]
AS 2.1560	Batch shake flasks	Cellobiose/xylose blend	25.5	52.0	Gong et al. [43]
GIM2.142	Batch shake flasks	Glucose/MSGWW *	4.6	24.7	Liu et al. [44]
CBS 1807	Batch shake flasks	Sweet sorghum	21.7	29.5	Matsakas et al. [45]
DSM 70296	Fed-batch bioreactor	Molasses	21.3	32.0	Vieira et al. [46]
DSM 70296	Fed-batch bioreactor	Sugar-cane bagasse	85.4	49.0	Anschau et al. [47]
DSM 70296	Fed-batch bioreactor	Flour-waste hydrolysate	109.8	57.8	Tsakona et al. [38]
3440 #	Fed-batch shake flasks	Glucose	18.0	40.5	Salunke et al. [48]
DSM 70296	Batch shake flasks	Crude glycerol	34.4	35.9	Tchakouteu et al. [35]
ATCC 56304	Batch shake flasks	Corn bran hydrolysate	23.5	33.3	Probst and Vadlani [49]
NRRL Y-11557	Batch shake flasks	Corn stover hydrolysate	24.6	38.7	Calvey et al. [50]
ATCC 56304	Fed-batch bioreactor	Glucose	81.6	41.8	Probst and Vadlani [51]
DSM 70296	Fed-batch bioreactor	SCGH **	87.4	46.0	Giannakis et al. [39]
DSM 70296	Batch shake flasks	Expired glucose	34.0	34.1	Present study

*: MSGWW is the monosodium glutamate waste-water; **: SCGH is the spent coffee grounds hydrolysate; #: Genetically engineered strain.

**Table 6 microorganisms-11-01863-t006:** Citric acid production by *Yarrowia lipolytica* strains growing on various carbon sources and fermentation configurations, and comparisons with the present investigation.

Strain	Culture Type	Substrate	CA (g/L)	Yield (g/g)	Reference
Wild-type strains					
NRRL Y-7576	Batch bioreactor	Glucose	51.5	0.71	Klasson et al. [58]
Y1095	Fed-batch bioreactor	Glucose	13.6–78.5	0.79	Rane and Sims [59]
ATCC 20346	Fed-batch bioreactor	Glucose	50–69	0.52	Moresi [60]
H222	Batch bioreactor	Glucose	62.0	0.37	Moeller et al. [61]
A-101	Batch bioreactor	Crude glycerol	66.8	0.43	Rywińska et al. [53]
H222	Fed-batch batch bioreactor	Glucose	97.7	0.56	Moeller et al. [55]
VKM Y 2373	Fed-batch bioreactor	Glucose	80–85	0.70–0.75	Kamzolova and Morgunov [62]
ACA YC 5029	Batch bioreactor	Crude glycerol	39.0	0.42	Papanikolaou et al. [19]
LMBF Y-46	Fed-batch bioreactor	Pure glycerol	101.3 ^#^	0.46	Papanikolaou et al. [57]
Mutants or genetically modified strains					
N1	Fed-batch bioreactor	Ethanol	120.0	0.85	Kamzolova et al. [63]
187/1	Fed-batch bioreactor	Rapeseed oil	135.0	1.55	Kamzolova et al. [54]
Wratislavia AWG7	Batch bioreactor	Crude glycerol	88.1	0.46	Rymowicz et al. [26]
H222-S4(p67ICL1)T5	Fed-batch bioreactor	Sucrose	133.0 ^#^	0.78	Förster et al. [64]
A-101-1.22	Fed-batch bioreactor	Crude glycerol	119.1 ^#^	0.64	Rymowicz et al. [28]
NG40/UV7	Fed-batch bioreactor	Pure glycerol	115.0	0.64	Morgunov et al. [56]
JMY 1203	Shake flasks	Crude glycerol	57.7 ^#^	0.91	Papanikolaou et al. [65]
NG40/UV5	Fed-batch bioreactor	Rapeseed oil	140.0	1.50	Morgunov et al. [66]
ACA-DC 50109	Fed-batch bioreactor	Expired glucose	82.0	0.50	Present study

^#^: Total citric acid (CA + *iso*-citric acid, in which *iso*-citric acid ranged between 6–12% of CA).

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
