# Peer review of "Growth Response of Non-Conventional Yeasts on Sugar-Rich Media: Part 1: High Production of Lipid by Lipomyces starkeyi and Citric Acid by Yarrowia lipolytica"

_microorganisms, 2023, doi:10.3390/microorganisms11071863_

Round 1
Reviewer 1 Report
The manuscript is interesting to read. I have some minor remarks mentioned below
- Please rewrite and organize the abstract according to the following context:
A short introduction, hypothesis (aim) of the study, methods, the most important quantitative results, a general conclusion, and future prospective.
- Conclusions section, please rewrite it with more detail and highlight the future standpoint well
All tables titles are long, please modify them. Other information might be moved to the table footer.
- Manuscript has grammatical errors, please check.
The manuscript has grammatical errors, please check.
Author Response
RESPONSES TO THE COMMENTS OF REFEREES
REFERENCE: Microorganisms-2514289
IMPORTANT NOTE: All changes, additions, modifications, etc, in the revised manuscript are found with yellow fond.
REFEREE 1
The manuscript is interesting to read. I have some minor remarks mentioned below:
- Please rewrite and organize the abstract according to the following context: A short introduction, hypothesis (aim) of the study, methods, the most important quantitative results, a general conclusion, and future prospective.
Response: The abstract was modified according to the suggestion of the referee. We have tried to reduce the length, but it was not evident, since small sections with “introduction”, “hypothesis” and “future prospective”, not existed in the previous version, were added.
- Conclusions section, please rewrite it with more detail and highlight the future standpoint well.
Response: The section “Concluding remarks” was significantly enriched, according to the referee’s suggestion.
All tables titles are long, please modify them. Other information might be moved to the table footer.
Response: Done, please see the paper.
- Manuscript has grammatical errors, please check.
Response: We have done our best to “polish” our paper.
REFEREE 2
The manuscript microorganisms-2514289 fits well the scope of Microorganism, and the results are informative for the field. However, the present version should be polished. Generally, the length is much too long, the expression in the manuscript should be refined.
- The “Abstract” is much too long. Please present the most important information in this part.
Response: The Abstract was modified (see also comment of Ref. 1). We have tried to reduce the length, but some items requested by Ref. 1 that did not previously exist were added.
- For all the tables, the details should better be placed under the tables.
Response: Done, please see the paper.
- For Fig.1, every picture should be labeled independently, please check throughout the manuscript.
Response: Done.
- For table 5 and table 6, please cite the most essential results, and over-citation should be avoided.
Response: Done, for both tables some lines were omitted. We have tried to maintain the most important information.
- Fig.2 should be improved.
Response: We do not understand how this TLC figure could be improved. We feel that this figure is rather informative. In any case, and if problems exist, we can directly omit this figure.
- For fig.1, the functions related "Global conversion yield of biomass produced per unit of glucose consumed (c) and global conversion yield of lipid consumed per unit of glucose consumed (d)" are confusing and difficult to understand.
Response: We have tried to explain in the figure legends (of both Fig. 1 and 3) the representation of the global conversion yield of biomass, lipid and citric acid produced per unit of glucose consumed. For Fig. 1, instead of performing the linear regression of produced biomass and lipid as function of the remaining glucose, we have changed and presented the linear regression of produced biomass and lipid as function of the consumed glucose, in order to render the graph more meaningful.
Reviewer 2 Report
Concerning Figure 1, identify the graphs as a, b, c, and d.
Reviewer 3 Report
The manuscript microorganisms-2514289 fits well the scope of Microorganism, and the results are informative for the field. However, the present version should be polished. Generally, the length is much too long, the expression in the manuscript should be refined.
1. the "Abstract" is much too long. Please present the most important information in this part.
2. For all the tables, the details should better be placed under the tables.
3. For Fig.1, every picture should be labeled independently, please check throughout the manuscript.
4. For table 5 and table 6, please cite the most essential results, and over-citation should be avoided.
5. Fig.2 should be improved.
6. For fig.1, the functions related "Global conversion yield of biomass produced per unit of glucose consumed (c) and global conversion yield of lipid consumed per unit of glucose consumed (d)" are confusing and difficult to understand.
Author Response

(The authors gave the same response as above.)
